# Functional Capacity Impairment in Long COVID After 17 Months of Severe Acute Disease

**DOI:** 10.3390/ijerph22020276

**Published:** 2025-02-13

**Authors:** Fernanda Facioli dos Reis Borges, Andrezza Cristina Barbosa Braga, Bernardo Silva Viana, Jefferson Valente, João Marcos Bemfica, Thaís Sant’Anna, Cássia da Luz Goulart, Fernando Almeida-Val, Guilherme Peixoto Tinoco Arêas

**Affiliations:** 1Human Movement Science Graduation Program, Universidade Federal do Amazonas, Manaus 6200, Brazil; fernanda.frb@hotmail.com; 2Physical Therapy Department, Universidade Federal do Amazonas, Manaus 6200, Brazil; andrezzab.braga@gmail.com (A.C.B.B.); thaissantanna@ufam.edu.br (T.S.); 3Medicine Department, Universidade Federal do Amazonas, Manaus 6200, Brazil; besilvaviana@gmail.com; 4Tropical Medicine Graduation Program, Universidade Estadual do Amazonas, Manaus 3578, Brazil; jefferson.valente@yahoo.com.br (J.V.);; 5Physical Therapy Department, Universidade de Brasília, Campus Ceilandia, Brasília 72220-275, Brazil; cassiadaluzgoulart@gmail.com; 6Tropical Medicine Foundation Dr. Heitor Vieira Dourado, Manaus 69040-000, Brazil; ffaval@gmail.com; 7Physiological Science Department, Universidade Federal do Amazonas, Manaus 6200, Brazil

**Keywords:** post-acute COVID-19 syndrome, walk test, functional capacity, oxygen consumption, quality of life

## Abstract

Long COVID represents a significant challenge in understanding the prolonged impact of the disease. Despite its increasing recognition, detailed insights into the long-term cardiopulmonary consequences remain sparse. This study aimed to evaluate the functional capacity of individuals with persistent symptoms after severe COVID-19 infection compared to control individuals without symptomatic COVID or mild COVID after 17 months. This is a case-control study assessing 34 individuals divided into two groups regarding functional capacity by distance in a 6-min walk test (D6MWT) associated with gas analysis, spirometry, respiratory muscle strength, and quality of life. During the 6 MWT, an important lower heart rate (HR) was observed for the COVID group (106 ± 10 bpm, difference mean: 21.3; *p* < 0.001), with greater exertional perception (Borg dyspnea: 4.5 [2.0–9.0], *p* < 0.001 and Borg fatigue: 4.0 [2.0–7.0], *p* = 0.01), a significant decrease in the distance covered (416 ± 94 m, difference mean: 107; *p* = 0.002), and a low value of O_2_ uptake ($\dot{V}$_O2_) (11 ± 5.0 mL/(kg min), difference mean: 8.3; *p* = 0.005) and minute ventilation (22 ± 8 L/min, difference mean: 18.6; *p* = 0.002), in addition to very low quality of life scores. Regression analysis showed a significant association between D6MWT and Borg fatigue and Borg dyspnea at rest (*p* = 0.003; *p* = 0.009). $\dot{V}$_O2_ and HR were also significantly associated with the outcomes of the D6MWT (*p* = 0.04 and *p* = 0.004, respectively). In conclusion, individuals who have severe COVID-19 and persist with symptoms have low functional capacity, low $\dot{V}$_O2_, low HR behavior, and low quality of life.

## 1. Introduction

Brazil was one of the countries hardest hit by the COVID-19 pandemic caused by SARS-CoV-2. The pandemic underscored the profound impact of pre-existing inequalities in the country. New risk scenarios, combined with the living and health conditions of the poorest groups, exacerbated already prevalent social vulnerabilities [1] (p. 78). This resulted in higher rates of mortality, lethality, incidence, and prevalence in the poorest regions, where the most vulnerable social groups are concentrated [2]. Within Brazil, Manaus emerged as a primary epicenter, reporting over 638,000 diagnosed cases and 14,000 official deaths [3]. Despite efforts at health management, the pandemic’s consequences remain evident in the lives of individuals affected by the acute phase of the illness.

COVID-19 infection is associated with multisystemic symptoms [4], which may persist beyond the acute phase—defined as extending more than four weeks after the initial infection [5]. This condition, known as post-COVID syndrome or long COVID [5], can affect individuals regardless of the severity of their initial illness, impacting both those with severe symptoms and asymptomatic cases [6]. Long COVID is highly heterogeneous, with varying definitions across studies. Currently, no universally accepted biomarkers or diagnostic tests are available [7]. Consistent risk factors for developing long COVID include female sex, older age, more severe acute infection, and lower socioeconomic status. Genetic and epigenetic factors may also contribute [7]. Although the precise mechanisms remain unclear, persistent symptoms are largely thought to stem from the systemic inflammatory response induced by SARS-CoV-2 during its acute phase. This leads to symptoms such as dyspnea, fatigue, reduced functional capacity, and diminished quality of life [8].

The presence and persistence of sequelae significantly increase morbidity and mortality among affected patients [9], emphasizing the vital role of rehabilitation in mitigating diverse symptoms and health complications. However, a comprehensive understanding of the pathophysiological changes associated with long COVID syndrome remains elusive. This highlights the urgent need for extended investigations and patient monitoring beyond the immediate pandemic crisis.

To date, research has identified physiological repercussions lasting up to six months post-infection, including impaired alveolar diffusion [10], altered autonomic behavior at rest and during both submaximal and maximal exercise, impaired vascular reactivity [11], reduced muscular strength [12], central nervous system damage, mitochondrial dysfunction in various organs, particularly muscles [13], and sustained pro-inflammatory activity indicative of impaired inflammatory control [4].

The persistence of post-COVID alterations is attributed to a combination of interconnected pathophysiological factors, particularly the prolonged systemic hyper-inflammatory response triggered by the viral infection. This ongoing inflammation affects multiple systems, including the hematologic, cardiac, renal, and vascular systems, contributing to sequelae such as endothelial dysfunction, thrombosis, and impaired perfusion of vital organs [9,14]. Direct organ injuries, including damage to the kidneys and heart caused by SARS-CoV-2, have also been linked to long-term functional loss, resulting in complications such as renal failure and cardiomyopathy [8,9,11].

Functional capacity is compromised in multiple ways due to these interconnected pathophysiological factors. Muscle injury can result from direct tissue damage, prolonged hyperinflammatory responses, and extended immobility, particularly in severely ill patients who have experienced lengthy hospitalizations. The ongoing inflammatory response further exacerbates skeletal muscle weakness and chronic fatigue [12]. Lung function is also significantly impaired by pathological processes such as destruction of the alveolar epithelium, hyaline membrane formation, capillary damage and bleeding, and fibrous proliferation in the alveolar septa. These changes can lead to pulmonary consolidation, vascular and alveolar remodeling, and the development of pulmonary fibrosis and/or chronic pulmonary hypertension [10,15], as well as restrictive and obstructive lung disorders [10]. However, the long-term persistence of these changes and their impact on functional capacity and $\dot{V}$_O2_ during capacity evaluations remain inconsistently documented.

The main research question of this study is: nearly two years into the pandemic, what is the level of impairment in the functional capacity of individuals who experienced severe COVID-19 and continue to report symptoms? This study aimed to evaluate the functional capacity of individuals with persistent symptoms following severe COVID-19 infection compared to control individuals without symptomatic COVID-19 or mild COVID-19 without fatigue. The main hypothesis is that individuals with post-severe COVID-19 symptoms experience significant impairments in functional capacity compared to those without symptoms.

## 2. Materials and Methods

### 2.1. Type of Study

The present study is an observational case-control study and adheres to the guidelines outlined in the Strengthening the Reporting of Observational Studies in Epidemiology (STROBE) statement. It was conducted in Manaus, Brazil, between April 2022 and June 2023. The study was approved by the Human Research Ethics Committee of the Universidade Federal do Amazonas (IRB: 64127022.5.0000.5020) and complied with the principles of the Declaration of Helsinki.

### 2.2. Population of Study

The study population consisted of two distinct groups. For the long COVID group, the inclusion criteria were individuals aged 30 to 65 years at the time of study participation, diagnosed with moderate to severe COVID-19 according to the World Health Organization (WHO) guidelines [5]. Mild cases were classified as: (1) flu-like syndrome with mild symptoms (no dyspnea or signs of severity, without evidence of viral pneumonia or hypoxia); (2) absence of decompensated comorbidities, without the need for hospitalization; and (3) home isolation for at least 10 days. Moderate cases were defined as: (1) moderate symptoms with clinical signs of pneumonia (fever, cough, dyspnea, rapid breathing requiring stabilization and admission to the ward, or non-invasive ventilatory support); (2) no signs of severe pneumonia, including SpO_2_ ≥ 90% on room air; (3) involvement of ≤50% of the lung parenchyma on computed tomography; (4) hospitalization of ≤10 days; and (5) respiratory and motor physiotherapy at least once a day. Severe cases were defined as requiring ICU admission with or without mechanical ventilation (MV) or non-invasive ventilation, pulmonary involvement of less than 50% of the lung parenchyma, and pneumonia-like signs such as SpO_2_ < 88% on room air.

A matched control group of individuals without long COVID consisted of subjects who had tested positive for COVID-19 but were asymptomatic or experienced mild COVID-19 and did not exhibit fatigue symptoms according to the Modified Medical Research Council (mMRC) scale. All participants presented a confirmatory polymerase chain reaction (RT-PCR) test for SARS-CoV-2 infection.

The following exclusion criteria were applied: participation in pulmonary rehabilitation within six months prior to the study; resting peripheral oxygen saturation (SpO_2_) lower than 88%; diagnosis of pulmonary diseases (e.g., COPD, asthma, pulmonary fibrosis) or heart diseases (e.g., chronic heart failure, post-transplant, post-myocardial infarction); presence of a pacemaker or implantable defibrillator; chronic degenerative, osteoarticular, or muscular dysfunctions that prevented the proposed assessments; decompensated oncological, metabolic, or renal diseases; uncontrolled systemic arterial hypertension; alcoholism; current or former smoking; history of illicit drug use; pregnancy; or claustrophobia related to wearing a mask during the walking test.

### 2.3. Recruitment

To reach individuals who met the eligibility criteria, informative posters were disseminated via social media and placed in high-traffic areas throughout the city. Additionally, contacts were made with participants from other studies involving COVID-19. After including participants in the long COVID group, individuals matched by sex, age, weight, and height were selected for the control group using the same recruitment strategies. After contacting participants, screening procedures were performed. Once the eligibility criteria were met, the participant’s assessment was scheduled.

### 2.4. Study Procedures

For all participants included in the study, the following procedures were performed: informed consent and signing of the informed consent form, assessment of medical history, clinical evaluation, spirometry, manovacuometry, quality-of-life questionnaire, and the 6 MWT associated with gas analysis. At the end of this assessment, an echocardiogram was also scheduled for the severe COVID-19 group.

#### 2.4.1. Interview

An interview was conducted to collect sociodemographic, epidemiological, and medical history data. The following variables are described: A history of COVID-19 infection with the date of diagnosis, date of hospitalization and discharge, the severity of infection, history of previous illnesses and comorbidities, medications in use, vital signs (blood pressure, heart rate, respiratory rate, body composition and weight, height, and BMI), SpO_2_, and fatigue assessed by the mMRC. The sensation of fatigue in daily life was assessed using the validated Portuguese version of the Modified Medical Research Council (mMRC) scale [16]. Following this interview, the following assessments were performed:

#### 2.4.2. Lung Function

Spirometry was performed using the Spirobank II portable device (MIR^®^, Rome, Italy) [17], according to the American Thoracic Society (ATS) guidelines [18]. The following variables were assessed: forced expiratory volume in one second (FEV1), forced vital capacity (FVC), FEV1/FVC ratio, and maximum voluntary ventilation (MVV). Reference values for the Brazilian population were used [19]. During the tests, specific disposable antibacterial/antiviral filters were used to prevent contamination of both the device and the participants.

#### 2.4.3. Ventilatory Muscle Strength

Manovacuometry was performed using the MVD300-U digital device (Globalmed^®^, São Paulo, Brazil) [20] to obtain the maximum inspiratory pressure (MIP) and maximum expiratory pressure (MEP). The tests followed the recommendations of the ATS/European Respiratory Society (ERS) [21] and the reference values described for the Brazilian population [22]. Antibacterial/antiviral filters were also used during the assessment.

#### 2.4.4. Quality of Life

To assess the quality of life (QoL), a Portuguese-validated version of the Short Form 36 (SF-36) was used [23]. The SF-36 is divided into eight scales with representative items. The score calculation followed the method recommended by Ware [24].

#### 2.4.5. Functional Capacity and Gas Analysis

The 6 MWT was used to assess the functional capacity of the participants and was performed following ATS recommendations [25]. Briefly, participants were instructed and encouraged to walk as far as possible in a 30 m-long flat corridor for a period of six minutes. The test was performed twice for each participant, with a 30-min interval between tests. The 6 MWTs were conducted in conjunction with a breath-by-breath ergospirometer (PNOE, Endo Medical^®^, Malvern, PA, USA) [26]. $\dot{V}$_O2_ (mL/(kg·min)), minute ventilation ($\dot{V}$_E_—L/min), and respiratory exchange rate (RER) were measured. In each test, the subject remained seated for six minutes, then rested for the same duration in an upright position, after which the 6 MWT began. Performance was recorded at the end of the exercise, along with the presence and duration of pauses during walking and any desaturation > 4%. A recovery time of six minutes after exercise was considered. A heart rate (HR) belt (Polar H10, Polar^®^, Kempele, Finland) [27], blood pressure (BP) monitor (HEM-7122, Omron^®^, Kyoto, Japan), peripheral SpO_2_ (model 2500, Nonin^®^, Plymouth, MN, USA), and the modified Borg CR-10 scale for subjective perception of exertion were assessed and recorded at rest, sitting pre-test, at the end of the test, and during the 6 min recovery period after the 6 MWT.

### 2.5. Statistical Analysis

The sample calculation was performed accepting the minimum clinically significant difference of 30 m walked in pulmonary disease [28] and an effect size (ES) of 0.85 Cohen’s D value. The following parameters were used: ES of 0.85 Cohen’s D value, α = 5%, β = 80%, and allocation of 1 for the unpaired *t*-student test, reaching a sample size of 46 volunteers. However, after reaching 10 participants in each group (n = 20), the sample calculation was remade, and values were found to be 520 ± 28 m for the control group and 480 ± 51 m for the severe COVID-19 group. Then, the ES was recalculated, and we arrived at 1.0 Cohen’s D value. Therefore, we used the following parameters as a final sample: ES of 1.0 of Cohen’s D value, α = 5%, β = 80%, and allocation of 1 for the unpaired *t*-student test, arriving at a final sample size of 34 volunteers, 17 into each group. However, the statistical power of the sample has not been calculated, nor has the experiment been previously designed to validate the representativeness of the sample.

Data are expressed as mean ± standard deviation (SD) of the mean, median, and interquartile range (IQR), percentage, difference between groups, and 95% confidence interval (CI 95%) for the difference between groups. Data normality was analyzed using the Shapiro–Wilk test and the Levene test for homogeneity. The unpaired t-student test was used for independent groups to analyze the difference between the means of the two groups, and the Mann–Whitney test was used to analyze the independent difference between the medians of the study groups. For categorical variables, the *x*^2^ test was used. To analyze the behavior of $\dot{V}$_O2_, $\dot{V}$_E_, and HR during the walking test ANOVA two-way test (between factors) post hoc Bonferroni test. For regression, the univariate method evaluated β, CI 95%, and r^2^. For regression, the multiple method was used, evaluating β and CI 95%, and to build the model values with *p*-value < 0.2 in bivariate methods, an autocorrelation *p*-value > 0.05, and the assumption of collinearity less than 5.0 was retained in the model. The *p*-value < 0.05 was accepted as statistically significant. To evaluate the ES, Cohen’s D value was used for normal data and the order biserial correlation for non-parametric data. The Jamovi 2.3 (Jamovi Project, Sydney, Australia), Graphpad Prism 8.0 (Graphpad, La Jolla, CA, USA), and G*Power 3.1 (University of Dusseldorf, Dusseldorf, Germany) software were used for analysis.

## 3. Results

In the end, 179 volunteers who had severe COVID-19 were screened for eligibility. Only 23 could be contacted. After screening, six volunteers were excluded (1 due to smoking, two due to uncontrolled hypertension, and three due to dropouts during the initial evaluation). In the control group, invitations were directly extended to volunteers matched for age, BMI, and sex, with all eligible volunteers selected for the study. A total of 34 individuals were included in the two groups (17 individuals in the severe COVID-19 group and 17 in the control group). Sample characteristics are shown in Table 1. The groups were matched by age, sex, and BMI. The majority were female, older than 46 years, with grade 1 obesity, normotensive, with SpO_2_ and resting HR within normal limits in both groups.

The majority of the COVID group had grade 2 dyspnea according to the mMRC scale, while the control group had grade 0 (*p* < 0.001). In the COVID group, the most common comorbidity was hypertension (41%), followed by anxiety and diabetes (18% each). Among the most used medications were AT1 blockers, followed by diuretics, anxiolytics, and hyperglycemic agents (18% each) (Table 1). Only two volunteers were former smokers, with one in each group. The volunteer in the COVID-19 group smoked for 7 years, 20 cigarettes per day, with a smoking history of 7 pack years. The volunteer in the control group smoked 20 cigarettes per day, with a smoking history of 2 pack years. Both volunteers are considered to have a light smoking history.

Lung function and respiratory muscle strength analyses showed a statistical difference between groups for the FEV1/FVC ratio (*p* < 0.001). Respiratory muscle strength and resistance were similar between both groups. Furthermore, no cardiac function abnormalities were found in the COVID group.

Important findings regarding the 6 MWT (Table 2) were observed. At rest, the severe COVID-19 group presented greater lower limb fatigue and dyspnea when compared with the control group (*p* < 0.05). During the test, a low HR was found in the severe COVID group in comparison to the control group (*p* < 0.001 with a large ES). The perception of exertion in the severe COVID-19 group at the end of the 6WMT was significantly elevated in comparison with the control group (*p* < 0.001). These behaviors were followed by a worse functional capacity in the Severe COVID-19 group, demonstrated by a significant decrease in the meters covered (control group: 523 ± 95 m; COVID group: 416 ± 94 m, *p* = 0.002) with less work performed during the test (control group: 40,490 ± 10,096 m · kg; Severe COVID-19: 32,626 ± 8023 m·kg, *p* = 0.019), with a large ES.

The RER was similar between both groups. A low $\dot{V}$_O2_ value was observed in the severe COVID-19 group compared to the control group (*p* = 0.005), accompanied by a low muscular efficiency verified by $\dot{V}$_O2_ in the COVID-19 compared to the control group (*p* = 0.009). In addition, a low ventilatory minute behavior was observed in the COVID group (*p* = 0.002).

Gas analysis during the 6 MWT showed that the $\dot{V}$_O2_, $\dot{V}$_E_, and HR were different. Figure 1A demonstrates the behavior of $\dot{V}$_O2_ during the 6 MWT between the groups, with a significant interaction (*p* < 0.001), with a significant difference between the groups after 20% of the test and remaining until the end of the test. Figure 1B demonstrates the behavior of the *V*_E_ during the test with a significant interaction (*p* = 0.0025) and a significant difference between the groups from 80% of the test until the end. Finally, Figure 1C demonstrates the behavior of HR during the test between groups, with a significant interaction (*p* < 0.0001), with the significant difference between groups appearing after 60% of the test.

The assessment of QoL showed a decrease in all parameters for the COVID group when compared to the control group. There was a large effect size for all variables. The functional capacity and the physical limitation variables had the highest different values when comparing both groups (*p* < 0.001) and with a large ES (Table 3). Univariate linear regression analysis showed a significant association between walked distance and Borg fatigue at rest (*p* = 0.003) and between walked distance and Borg dyspnea at rest (*p* = 0.009). $\dot{V}$_O2_ and HR were also significantly associated with the outcomes of the 6 MWT (*p* = 0.04 and *p* = 0.004, respectively).

The Borg fatigue and the Borg dyspnea assessment explained 43% and 30% of the walked distance, respectively. The $\dot{V}$_O2_ and HR explained 16% and 25% of the 6 MWD, respectively, in the regression between the 6 MWD and QoL assessment. The functional capacity, physical limitation, pain, vitality, and mental health were all significantly associated (*p* < 0.01) with the 6 MWD. These variables explained 34%, 38%, 25%, and 32% of the outcomes of the 6 MWT (Table 4).

In the multiple method linear regression showed a significant influence of $\dot{V}$_O2_ (mL/(kg·min)), Borg fatigue in the rest, functional capacity, physical limitation, vitality, and mental health on the 6 MWD and explained 91% of 6 MWD (values of model: r = 0.955; r^2^ = 0.912; F = 13.2; *p* < 0.001) (Table 5).

## 4. Discussion

This study is the first to evaluate physical capacity using gas analysis during the 6 MWT in patients with severe COVID-19 who continue to experience significant effort perception (grade 2 on the mMRC scale). Our findings highlight a substantial and persistent decline in functional capacity among patients dealing with long-term COVID-19 compared to those unaffected by the condition. These patients exhibited reduced $\dot{V}$_O2_, decreased ventilatory activity, an attenuated heart rate response, and lower quality of life compared to the control group. Remarkably, this functional impairment persists nearly two years after the acute illness despite the absence of pulmonary dysfunction or impairments in ventilatory muscle strength and endurance.

The persistence of multisystemic symptoms and functional limitations associated with long-term COVID-19 in individuals who experienced severe COVID-19 has been consistently demonstrated in numerous studies [4,8,9,10,12,29,30,31,32,33,34]. While evidence shows that symptom persistence can occur in both severe and mild cases [6], disease severity is considered a significant risk factor for long COVID [7]. Pre-existing comorbidities before COVID-19 pose challenges, particularly in distinguishing between new conditions following infection, the “unmasking” of pre-clinical or subclinical conditions [7], or the exacerbation of pre-existing conditions by long COVID [6].

There remains room to expand evidence on assessing functional capacity using field tests combined with gas analysis in this population, with the current study being the first to demonstrate such findings. Furthermore, most studies investigating long COVID have focused on evaluations up to 6 months [30,31] or 12 months [32,33] post-acute infection. Few studies have explored functional capacity and quality of life (QoL) in this population beyond the 26 ± 9 months mark since the onset of the acute illness.

Xie and Al-Aly [9] conducted a comprehensive 2-year follow-up study examining the prevalence of 80 multisystemic sequelae and their association with risks of death and hospitalization. Notably, after this period, 35% of these sequelae decreased among previously hospitalized patients. The authors emphasize that while the risk of mortality may have lessened, the significant impact of long COVID on health necessitates ongoing attention to these individuals’ well-being.

In a 12-month follow-up study, lung function and QoL were evaluated in 46 individuals previously hospitalized for COVID-19. Using spirometry, whole-body plethysmography, and carbon monoxide diffusion tests, researchers observed a restrictive pattern in 28% of patients. Notably, those with severe COVID-19 demonstrated reduced diffusing capacity for carbon monoxide (DLCO). Additionally, the study revealed a direct correlation between disease severity and the extent of QoL impairment [29]. Our findings corroborate these results.

Interestingly, Patton et al. [34] showed that patients with more severe symptoms, particularly fatigue and cough, exhibited lower DLCO values. In 18% of volunteers with DLCO dysfunction, no restrictive lung patterns were detected via plethysmography. In our study, spirometry was used to assess lung function. However, parallels with Patton et al.’s findings suggest a relationship between low $\dot{V}$_O2_ during functional assessment and reduced oxygen utilization efficiency by peripheral muscles, as indicated by the ratio of distance walked to $\dot{V}$_O2_. This implies potential impairments in pulmonary and peripheral oxygen diffusion.

Complementing Patton et al. [34], Tryfonos et al. [35] demonstrated the effects of different exercise types (high intensity, moderate intensity, and strength training) on post-exercise peripheral muscle alterations. They observed significant musculoskeletal dysfunction after exercise, including fatigue and joint pain, compared to the control group. Charlton et al. [36] demonstrated that chronic fatigue caused by long COVID is directly related to significant mitochondrial dysfunction, which has also been associated with the recently identified mechanism of external malaise. Alongside metabolic changes, alterations in capillary density, endothelial dysfunction, muscle fiber type distribution, autoimmune changes [36,37], and abnormalities in the central and peripheral nervous systems [36] have been reported.

These findings highlight severe post-COVID-19 myopathy, even though the disease severity of study participants was not specified. This suggests that combined pulmonary (alveolar diffusion disorders), musculoskeletal impairments, and changes in the central nervous system may contribute to the persistent decline in functional capacity long after acute infection.

Dyspnea and fatigue are predominant symptoms reported by individuals with long-term COVID-19, with respiratory muscle weakness emerging as a potential underlying cause [12]. Investigations into inspiratory muscle strength during COVID-19 hospitalization and up to one month post-discharge revealed notable declines at various stages, including ICU admission, hospital discharge, and one month post-discharge. Additionally, these studies showed a significant correlation between maximal sustained inspiratory efforts and the presence of dyspnea [30]. In our study, no decreases in respiratory muscle strength or ventilatory resistance were observed despite reductions in $\dot{V}$_E_ during the 6 MWT. This discrepancy could be attributed to lower respiratory muscle resistance, raising the question of whether MVV is the most appropriate method for assessing ventilatory muscle resistance in this population, warranting further research.

A Turkish study [31] similarly assessed lung function via spirometry and identified a restrictive pattern in 21.5% of participants. Significant statistical differences were found in FEV1 and FVC values between severely and moderately ill individuals. Functional capacity was assessed using the 6 MWT without predicted distance calculation. The study reported that a walking distance of less than 427 m was abnormal. Participants with a restrictive spirometry pattern averaged 390 m, while those with normal spirometry averaged 430 m. Our findings align with an average distance of 416 m among participants without a restrictive pattern.

We also observed lower HR responses among patients, with insufficient compensatory adjustments to increased functional activity. A systematic review with meta-analysis incorporating nine studies and 464 participants similarly concluded that these patients exhibit signs of deconditioning and peripheral limitations, including abnormal oxygen uptake, dysfunctional breathing, and chronotropic incompetence [32]. These assessments were conducted using cardiopulmonary testing. A French study conducted cardiopulmonary tests 12 months post-COVID-19 hospitalization [33] found $\dot{V}$_O2_ values within normal limits in 80% of the sample but reported ventilatory inefficiency, consistent with the $\dot{V}$_E_ behavior observed in our study.

A significant factor potentially worsening patients’ health is their notably low QoL. Numerous studies have established a clear link between symptom persistence and reduced QoL among COVID-19 recovery patients [38,39]. Evidence also suggests a correlation between emotional well-being and other QoL domains, indicating that an individual’s health perception is significantly influenced by their mental state [40].

### 4.1. Future Perspectives

Future studies could expand on long-term cohort evaluations to determine how reduced functional capacity impacts not only QoL but also hospitalization rates and negative outcomes such as mortality. While studies highlight the importance of physical rehabilitation in post-COVID-19 patients [41,42], limited data exist for individuals recovering from severe acute illness sequels. Questions such as whether rehabilitation persists in improving functional capacity, whether the functional gains are consistent with those in non-COVID-19 sequelae, and whether post-exercise fatigue is proportionally greater in varying severities remain to be answered.

### 4.2. Limitation

We have some limitations in the study. The number of participants complicates the external validation of the study. However, using sample size calculations, the chance of systematic errors is much lower, with values below 5% for the most important variables. Additionally, these findings allow treatment to be directed toward the identified changes to try to alleviate or minimize the persistent symptoms that affect patients’ lives in so many ways. We also recommend that similar studies be replicated in a larger number of patients to confirm our findings. Lastly, the difficulty in knowing the pre-existing conditions related to comorbidities limits further inferences about the physical disability of the volunteers. However, based on other studies that support us, we believe these limitations have greater effects on the sequelae of COVID-19.

## 5. Conclusions

Our study evaluated the functional capacity of 17 individuals who had severe COVID-19 and compared it with a matched control group; we found important maintenance of functional changes after all this time, mainly related to a poor adaptation to functional capacity assessment and the low-quality of life assessed on these individuals. Thus, we conclude that individuals who had severe COVID-19 and persist due to long COVID have important low functional capacity, low capacity to mobilize gases, low muscle quality, low HR behavior, and a very low QoL compared to subjects in the control group and did not have severe COVID-19. We reaffirm that these findings allow treatment to be directed toward the identified changes to try to alleviate or minimize the persistent symptoms that affect patients’ lives in so many ways. Furthermore, it is important to conduct studies that continue to longitudinally observe how these volunteers will be affected by low functional capacity and whether they undergo rehabilitation or not. We also recommend that similar studies be replicated in a larger number of patients to confirm our findings.

## Figures and Tables

**Figure 1 ijerph-22-00276-f001:**
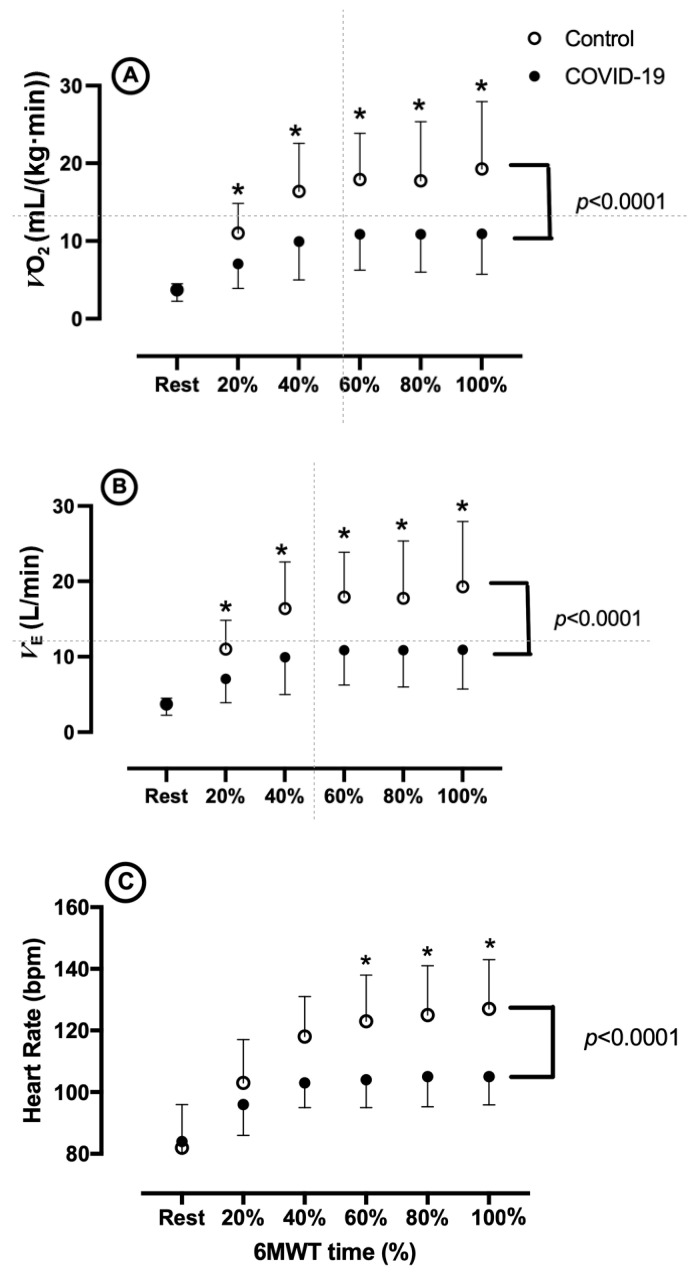
(**A**) $\dot{V}$_O2_ behavior during the 6-min walk test between groups. (**B**) $\dot{V}$_E_ behavior during the 6-min walk test between groups. (**C**) HR behavior during the 6-min walk test between groups. * *p* < 0.05 Control vs. Severe COVID-19.

**Table 1 ijerph-22-00276-t001:** Baseline characteristics of groups.

	Control Group (n = 17)	Severe COVID-19 (n = 17)	*p*-Value
Age (years)	45 ± 8	46 ± 8	0.59
Sex (Male/Female) [n]	3/14	3/14	0.99
BMI (kg/m^2^)	31 ± 6	32 ± 7	0.47
mMRC	0.0 [0.0–1.0]	2.0 [2.0–3.0]	<0.001
SBP (mmHg)	120 ± 15	120 ± 16	0.86
DBP (mmHg)	81 ± 9.3	84 ± 11	0.52
HR (bpm)	79 ± 9.5	84 ± 9.3	0.13
SpO_2_ (%)	97 ± 1.3	98 ± 2.1	0.50
Lung function			
FVC (L)	3.4 ± 0.8	3.4 ± 0.6	0.97
FVC (%)	99 ± 13	93 ± 12	0.21
FEV_1_ (L)	2.8 ± 0.5	2.7 ± 0.4	0.41
FEV_1_ (%)	105 ± 23	93 ± 15	0.10
FEV_1_/FVC	92 ± 10	79 ± 8	<0.001
Respiratory Muscle Strength			
MVV (L/min)	108 ± 38	96 ± 38	0.35
MVV (%)	83 ± 23	75 ± 28	0.45
MIP (cmH_2_O)	82 ± 32	83 ± 27	0.86
MIP (%)	83 ± 31	88 ± 32	0.65
MEP (cmH_2_O)	98 ± 41	96 ± 32	0.88
MEP (%)	104 ± 43	98 ± 32	0.57
Diagnosis time (months)	-	26 ± 9.0	-
Hospitalized	-	90%	-
Hospitalized in 2020	-	41%	-
Hospitalized in 2021	-	59%	-
Comorbidities			
Anxiety	0.0%	18%	-
Hypertension	12%	41%	0.009
Diabetes	0.0%	18%	-
Drugs			
AT1 blocker	12%	29%	0.001
Beta blocker	0.0%	12%	-
Diuretic	0.0%	18%	-
ACE inhibitor	0.0%	5%	-
Anxiolytic	0.0%	18%	-
Hyperglycemic	0.0%	18%	-

BMI: body mass index; mMRC: modified Medical Research Council; SBP: systolic blood pressure, DBP: diastolic blood pressure; HR: heart rate; SpO_2_: peripheral oxygen saturation; FVC: forced vital capacity; FEV_1_: forced expiratory volume in the first second; MVV: maximum voluntary ventilation; MIP: maximum inspiratory pressure; MEP: maximum expiratory pressure; AT1: type I angiotensin II receptor; ACE: angiotensin-converting enzyme.

**Table 2 ijerph-22-00276-t002:** 6 MWT analysis and gas analysis.

	Control Group (n = 17)	Severe COVID-19 Group (n = 17)	Diff Value	CI 95%	*p*-Value	ES (CI 95%)
Rest						
HR (bpm)	81 ± 14	85 ± 8	−0.4	−6.8–6.0	0.89	−0.04 (−0.7–0.3)
SBP (mmHg)	120 ± 22	120 ± 19	−0.7	−15–12	0.92	−0.03 (−0.7–0.6)
DBP (mmHg)	84 ± 13	82 ± 14	1.5	−7.9–11	0.74	0.1 (−0.5–0.7)
SpO_2_ (%)	97 ± 1.3	97 ± 1.6	0.1	0.05–0.25	0.50	−0.09 (−0.2–0.4)
Borg fatigue	0.0 [0.0–2.0]	2.0 [0.0–5.0]	−2.0	−2.0–−1.0	<0.001	0.63 *
Borg dyspnea	0.0 [0.0–2.0]	2.0 [0.0–7.0]	−2.0	−3.0–−0.4	0.003	0.56 *
Peak exercise						
HR (bpm)	127 ± 16	106 ± 10	21.3	10–32	<0.001	1.6 (0.7–2.4)
SBP (mmHg)	131 ± 21	128 ± 20	2.8	−11–17	0.69	0.1 (−0.5–0.8)
DBP (mmHg)	86 ± 10	86 ± 13	0.1	−8.1–8.5	0.97	0.01 (−0.6–0.6)
SpO_2_ (%)	98 ± 1.2	98 ± 1.9	0.2	−0.4–3.1	0.89	0.04 (−0.6–0.3)
Borg fatigue	1.0 [0.0–5.0]	4.0 [2.0–7.0]	−3.0	−4.0–−1.0	<0.001	0.72 *
Borg dyspnea	2.0 [0.0–5.0]	4.5 [2.0–9.0]	−2.9	−3.9–1.0	<0.001	0.69 *
Distance (m)	523 ± 95	416 ± 94	107	40–173	0.002	1.1 (0.3–1.8)
Distance (%)	92 ± 16	74 ± 18	18	5.9–30	0.005	1.0 (0.3–1.8)
Work (m·kg)	40,490 ± 10,096	32,626 ± 8023	7864	1362–14,365	0.019	0.8 (0.2–1.2)
Gas analysis						
$\dot{V}$_O2_ (mL/(kg·min))	19 ± 9.0	11 ± 5.0	8.3	2.3–13	0.005	1.2 (0.4–1.9)
RER	0.91 ± 0.1	0.95 ± 0.2	−0.4	−0.2–0.1	0.62	0.2 (0.08–0.7)
$\dot{V}$_E_ peak (L/min)	40 ± 16	22 ± 8.0	18.6	7.8–29	0.002	1.4 (0.6–2.1)
m/$\dot{V}$_O2_ (m/(mL/(kg·min)))	20 ± 8.20	51 ± 36	−31	−54–−8.4	0.009	1.1 (0.4–1.8)

ES: Effect size; HR: heart rate; SBP: systolic blood pressure; DBP: diastolic blood pressure; SpO_2_: peripheral oxygen saturation; $\dot{V}$_O2_: oxygen uptake, RER: respiratory exchange ration; $\dot{V}$_E_: minute volume. * Order biserial correlation effect size.

**Table 3 ijerph-22-00276-t003:** Quality of life between groups by SF-36.

	Control Group (n = 17)	Severe COVID-19 Group (n = 17)	Diff value	CI 95%	*p*-Value	ES (CI 95%)
Functional capacity	82 ± 15	38 ± 19	44	31–57	<0.001	2.5 (1.2–3.7)
Physical limitation	84 ± 29	10 ± 21	74	55–93	<0.001	2.9 (1.5–4.2)
Pain	66 ± 13	35 ± 14	31	17–44	<0.001	1.7 (0.7–2.7)
General state	65 ± 19	44 ± 22	21	6.3–35	0.007	1.0 (0.2–1.9)
Vitality	64 ± 20	31 ± 16	33	20–47	<0.001	1.8 (0.8–2.8)
Social aspects	74 ± 26	40 ± 22	34	16–52	<0.001	1.4 (0.5–2.3)
Emotional limitations	69 ± 39	31 ± 41	37	7.1–68	0.018	0.9 (0.1–1.7)
Mental health	72 ± 17	50 ± 15	21	9.3–34	0.002	1.3 (0.4–2.2)

CI 95%: Confidence interval 95%, Effects Size: ES.

**Table 4 ijerph-22-00276-t004:** Univariate regression between walked distance at the 6 MWT, Borg at rest, oxygen uptake, ventilation, heart rate, and SF-36 variables.

	*β*	CI 95%	r	r^2^	*p*-Value
Borg Fatigue rest					
5–0	−351	−540–−161	0.66	0.43	<0.001
Borg Dyspnea rest					
5–0	−204	−352–−57	0.55	0.30	0.008
$\dot{V}$_O2_ (mL/(kg min))	5.92	0.04–11	0.39	0.16	0.042
HR (bpm)	2.24	0.7–3.7	0.50	0.25	0.004
Functional capacity	2.25	1.0–3.4	0.58	0.34	<0.001
Physical limitation	1.61	0.9–2.3	0.66	0.44	<0.001
Pain	0.51	0.2–0.8	0.61	0.38	<0.001
Vitality	2.24	0.7–3.7	0.50	0.25	0.004
Mental health	3.20	1.4–4.9	0.57	0.32	<0.001

$\dot{V}$_O2_: oxygen uptake, HR: heart rate, CI 95%: Confidence interval 95%.

**Table 5 ijerph-22-00276-t005:** Multiple linear regression to evaluate the influence of variables on walked distance at the 6 MWT.

	*β*	CI 95%	*p*-Value
$\dot{V}$_O2_ (mL/(kg·min))	−2.9	−6.1–0.1	0.063
Borg Fatigue rest			
1–0	241	132–350	<0.001
5–0	−330	−445–−215	<0.001
Functional capacity	1.3	−0.04–2.6	0.058
Physical limitation	1.1	0.3–1.8	0.006
Vitality	−2.0	−3.3–−0.6	0.006
Mental health	1.6	0.1–3.1	0.037

$\dot{V}$_O2_: oxygen uptake, CI 95%: Confidence interval 95%.

## Data Availability

The original contributions presented in the study are included in the article. The raw data supporting the conclusions of this article shall be made available upon reasonable request to the corresponding author.

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
