# Peer review of "Functional Capacity Impairment in Long COVID After 17 Months of Severe Acute Disease"

_ijerph, 2025, doi:10.3390/ijerph22020276_

Round 1
Reviewer 1 Report
Comments and Suggestions for Authors
Thank you for sending this contribution.
In general, the paper is good, but I would like to point out some considerations.
- The first sentence of the introduction (lines 49-50) I think is incomplete without explaining that the main cause of the great affection that Brazil suffered is due to the non-action policies taken by the government.
- The statistical power of the sample has not been calculated, nor has the experiment been previously designed to validate the representativeness of the sample. It should be added in the methodology.
- The worst part of the study methodology is that there was no study of the characteristics of the subjects before suffering the disease. This does not invalidate the results, but we already knew previously that precisely the subjects with the worst habits (no exercise, smoking, etc.) had the worst prognosis. This is clearly seen in Table 1 when we see the comorbidities that we do not know if they were previously present. This part should have been discussed.
Author Response
Thank you for sending this contribution.
In general, the paper is good, but I would like to point out some considerations.
- The first sentence of the introduction (lines 49-50) I think is incomplete without explaining that the main cause of the great affection that Brazil suffered is due to the non-action policies taken by the government.
Answer: Thank you very much for the suggestion. Unfortunately, Brazil faced a rather unique situation in addressing COVID, particularly in terms of government response. However, we prefer not to include this information in the text, as we experienced personal challenges due to COVID outcomes, and there is still some backlash when political matters are mentioned. For this reason, we chose not to reference political situations in the introduction.
- The statistical power of the sample has not been calculated, nor has the experiment been previously designed to validate the representativeness of the sample. It should be added in the methodology.
Answer: Thank you very much for the suggestion. We have added this information in the statistics section, lines 224 to 225.
- The worst part of the study methodology is that there was no study of the characteristics of the subjects before suffering the disease. This does not invalidate the results, but we already knew previously that precisely the subjects with the worst habits (no exercise, smoking, etc.) had the worst prognosis. This is clearly seen in Table 1 when we see the comorbidities that we do not know if they were previously present. This part should have been discussed.
Answer: Thank you very much for the suggestion. Since we did not have pre-pandemic data on these individuals, we were unable to separate the effects of the pre-existing condition from the current condition. We have included this information in the discussion, on lines 381 to 387. In addition, we have included the information in the study limitations section.
Reviewer 2 Report
Comments and Suggestions for Authors
Dear Editor,
Thank you very much for the opportunity to review the paper entitled “EXERCISE INTOLERANCE IN LONG COVID AFTER 17 2 MONTHS FROM SEVERE ACUTE DISEASE”.
Although the research is interesting, I believe the title is misleading because exercise is used to evaluate variables like HR, etc.
The authors had to write about other post-Covid research in the introductory section as well as the gap that this study fill in the literature
Many post-Covid studies have been mentioned in the conclusion. That needs to be altered.
Methodology
I'd like to know from the authors how many men and women participate in the study and whether there were any differences.
The participants' ages vary in the section materials and in Table 1.
Why are just the females mentioned in Table 1?
An appendix should provide a description of the SF-36 version.
The conclusion should be rewritten using the findings of your study rather than those of previous studies.
Comments on the Quality of English Language
Quality of English is good.
Author Response
Review 2
Dear Editor,
Thank you very much for the opportunity to review the paper entitled “EXERCISE INTOLERANCE IN LONG COVID AFTER 17 MONTHS FROM SEVERE ACUTE DISEASE”.
Q: Although the research is interesting, I believe the title is misleading because exercise is used to evaluate variables like HR, etc.
Ans: I appreciate your comment and for finding the study interesting. You’re right; the title is incorrect regarding exercise intolerance. We will revise it to: “Functional Capacity Impairment in Long COVID After 17 Months from Severe Acute Disease.”
Q: The authors had to write about other post-Covid research in the introductory section as well as the gap that this study fill in the literature. Many post-Covid studies have been mentioned in the conclusion. That needs to be altered.
Ans: We have included new studies on Long COVID in the discussion, as indicated.
Q: Methodology
I'd like to know from the authors how many men and women participate in the study and whether there were any differences.
Ans: Thank you very much for the opportunity to discuss this topic. Perhaps it was not clear in the criteria, and we have clarified this in the text, but we included both sexes in the study. At the time the study was conducted, there was limited information about sex differences in long COVID. However, we now know that there is a difference in physical capacity between men and women in patients with long COVID. A study published by our group, in collaboration with other centers, demonstrated differences between sexes in physical capacity and other variables during cardiopulmonary testing (Goulart et al., Sex-Based Differences in Pulmonary Function and Cardiopulmonary Response 30 Months Post-COVID-19: A Brazilian Multicentric Study. International Journal of Environmental Research and Public Health, 2024. https://www.mdpi.com/1660-4601/21/10/1293). In the current study, conducting a sex-specific analysis of functional capacity would have resulted in a much smaller sample size and a smaller effect size for detecting these differences. Furthermore, the study was matched by sex, age, weight, and height (and BMI), so we believe that in this study, sex does not influence the outcomes.
Q: The participants' ages vary in the section materials and in Table 1.
Ans: We put in the material section the inclusion criteria and improve the specified age range. In the table, we had the mean age and standard deviation, which aligned with the description in the materials and methods.
Q: Why are just the females mentioned in Table 1?
Ans: I think we didn't make it very clear; we have adjusted.
Q: An appendix should provide a description of the SF-36 version.
Ans: I appreciate the feedback; however, the SF-36 is not a questionnaire produced by our group. I believe that including the document as a supplement may cause copyright issues. We are concerned about potential problems with the authors since we have already cited it in our reference.
Q: The conclusion should be rewritten using the findings of your study rather than those of previous studies.
Ans: Thank you, it has been expanded.
Reviewer 3 Report
Comments and Suggestions for Authors
Dear Authors,
I would like to express my gratitude for the opportunity to review this manuscript.
At this stage, the document requires improvements, below with lines indication:
2-3 – Please correct the title format considering the journal template and instructions for authors.
3-23 – Please correct the affiliations format.
35-38 – Please indicate VO2 in full before abbreviating. Moreover, VO2 is different from VO2max.
34-39 – Please consider presenting numerical results.
53-54 – Please revise the citations format throughout the manuscript.
117, 121 – Please revise the subtitles format (e.g. – space before and after subtitles; upper and lowercase in the subtitles).
122-136 – Please consider more clearly presenting the inclusion and exclusion criteria.
122-149 – Please indicate all information’s related to the subjects. Some examples, past sporting experience, training routines, medicine? Previous injuries? Familiarization with the procedures? Warm-up description. These are only some examples.
158 – Please address the ethical issues.
176 – Considering all the instruments, please indicate manufacturer, version, city and country. Additionally, please consider references to support the procedures.
202 – Please consider middle dots (e.g. “ml.kg.min“).
222-225 – If sample power was not used why mention GPower in this section? Please revise in detail the statistical analysis section. For example, the ES should be described (with the presentation of numerical intervals) and supported by reference.
256 – “p” in italics, please revise all manuscript.
269 & 288 – Please revise the variables values (e.g. age SD), present all the units, and revise the table footnote, considering the journal template. Moreover, the tables format (e.g. space between title and table) should be revised in detail considering the journal template.
308 – Please revise the units’ format (middle dots, for example).
346 – “ES” is presented many times in the manuscript, please consider abbreviating in the first appearance, and afterward, only presenting the abbreviation.
348 – Please revise the decimals. Additionally, the p values are in bold in other tables, but not in this one, please standardize. Moreover, the ES intervals should be described in the statistical analysis section and in the table footnote.
362 & 369 – Please revise the tables´ content and format.
372 – Please consider improving the discussion section quality and rationale. Also, please consider standardizing the paragraphs size to improve readability.
466 – Please indicate suggestions for future research.
506 – Please revise and correct all the references format, considering standardization assuming the journal template.
Please carefully revise the English in the manuscript.
Comments on the Quality of English LanguagePlease revise and improve the English quality.
Author Response
Review 3
Dear Authors,
I would like to express my gratitude for the opportunity to review this manuscript.
Ans: We are the ones who thank you for your review, with sincere gratitude for improving our manuscript. It was very easy to organize the corrections by following the line.
At this stage, the document requires improvements, below with lines indication:
Q: 2-3 – Please correct the title format considering the journal template and instructions for authors.
Ans: Thank you, we have already corrected it.
Q: 3-23 – Please correct the affiliations format.
Ans: Thank you, we have already corrected it.
Q: 35-38 – Please indicate VO2 in full before abbreviating. Moreover, VO2 is different from VO2max.
Ans: Thank you, we have already corrected it. Indeed, VO2max is different from VO2, and this has been modified throughout the manuscript.
Q: 34-39 – Please consider presenting numerical results.
Ans: Thank you, we have already corrected it.
Q: 53-54 – Please revise the citations format throughout the manuscript.
Ans: Thank you, we have already corrected it.
Q: 117, 121 – Please revise the subtitles format (e.g. – space before and after subtitles; upper and lowercase in the subtitles).
Ans: Thank you, we have already corrected it.
Q: 122-136 – Please consider more clearly presenting the inclusion and exclusion criteria.
Ans: Thank you, we have already adjusted and added the criteria to make it clearer.
Q: 122-149 – Please indicate all information’s related to the subjects. Some examples, past sporting experience, training routines, medicine? Previous injuries? Familiarization with the procedures? Warm-up description. These are only some examples.
Ans: All volunteers were sedentary and had no previous involvement with physical activity or exercise. Since the inclusion criteria also specified sedentary volunteers, questionnaires like the IPAQ, which assess only the year prior to the current evaluation, would not be able to deeply assess the volunteers past physical activity. We appreciate your question.
Q: 158 – Please address the ethical issues.
Ans: Thank you, we have already corrected it.
Q: 176 – Considering all the instruments, please indicate manufacturer, version, city and country. Additionally, please consider references to support the procedures.
Ans: Thank you, we have already corrected it.
Q: 202 – Please consider middle dots (e.g. “ml.kg.min“).
Ans: Thank you, we have already corrected it.
Q: 222-225 – If sample power was not used why mention GPower in this section? Please revise in detail the statistical analysis section. For example, the ES should be described (with the presentation of numerical intervals) and supported by reference.
Ans: G*Power was used to calculate the sample size at all stages of the study, which is why it was mentioned. The confidence interval for the reported effect size was included.
Q: 256 – “p” in italics, please revise all manuscript.
Ans: Thank you, we have already corrected it.
Q: 269 & 288 – Please revise the variables values (e.g. age SD), present all the units, and revise the table footnote, considering the journal template. Moreover, the tables format (e.g. space between title and table) should be revised in detail considering the journal template.
Ans: Thank you, we have already corrected it.
Q: 308 – Please revise the units’ format (middle dots, for example).
Ans: Thank you, we have already corrected it.
Q: 346 – “ES” is presented many times in the manuscript, please consider abbreviating in the first appearance, and afterward, only presenting the abbreviation.
Ans: Thank you, we have already corrected it.
Q: 348 – Please revise the decimals. Additionally, the p values are in bold in other tables, but not in this one, please standardize. Moreover, the ES intervals should be described in the statistical analysis section and in the table footnote.
Ans: The p-values in bold were only for those below 0.05. However, we removed them to avoid any issues. Additionally, we have added the statistical description.
Q: 362 & 369 – Please revise the tables´ content and format.
Ans: Thank you, we have already corrected it.
Q: 372 – Please consider improving the discussion section quality and rationale. Also, please consider standardizing the paragraphs size to improve readability.
Ans: Thank you, it has been expanded.
Q: 466 – Please indicate suggestions for future research.
Ans: Excellent idea, we did not include a discussion on future research and perspectives. Thank you for the suggestion, and we have added it.
Q: 506 – Please revise and correct all the references format, considering standardization assuming the journal template.
Ans: Thank you, it has been expanded.
Q: Please carefully revise the English in the manuscript.
Ans: We had reviewed it earlier. After the insertion, it was reviewed again. But thank you for your comments.
Round 2
Reviewer 1 Report
Comments and Suggestions for Authors
Thank you for taking the time to review the issues noted in the previous review. We believe that the research is now ready for publication. There are still some points that could have been improved and others that methodologically could not be improved at such an advanced stage. However, we consider that in general the result is good and we congratulate them for it.
Reviewer 3 Report
Comments and Suggestions for Authors
Dear Authors,
Thank you for the work performed in the manuscript.
The manuscript still requires deeply improvement, below some suggestions:
Title – The title should present upper and lowercase. Please revise and consider the journal template and instructions for authors.
47-111 – The letter and spaces format do not seem according to the journal template, please revise.
51 – All citations formats should be revised and corrected considering the journal template and instructions for authors.
186-197 – Different subtitles format, please revise and correct format details in all manuscript.
197-206 – Please correct the numeration.
263 – Please remove the comment.
288 – “p” in italics should be present throughout the manuscript.
311 – “p-valor” – please change to English. Same in table 2. Please also revise the tables’ content aiming for a more clear and direct presentation to the readers. For example, table 1 crosses two pages.
371 – Please consider improving the figure quality.
400 – Please revise the table content and format. Please consider this in all tables (1 to 5).
The discussion and following sections quality of the writing can be improved.
All references should be corrected, according to the journal template and instructions for authors.
Comments on the Quality of English LanguageEnglish details can be improved.
Author Response
Dear Authors,
Thank you for the work performed in the manuscript.
Ans: I appreciate the careful review of the study. Thank you for your feedback.
The manuscript still requires significant improvement. Below are some suggestions:
Title – The title should use both uppercase and lowercase letters. Please revise it in accordance with the journal template and instructions for authors.
Ans: Adjusted. Thank you for pointing that out.
47-111 – The formatting of letters and spaces does not seem to comply with the journal template. Please revise.
Ans: Adjusted. Thank you for the suggestion.
51 – All citation formats should be revised and corrected according to the journal template and instructions for authors.
Ans: Adjusted. Thank you for pointing that out.
186-197 – Different subtitle formats are present. Please revise and ensure formatting consistency throughout the manuscript.
Ans: Adjusted. Thank you for your feedback.
197-206 – Please correct the numbering.
Ans: Adjusted. Thank you for the suggestion.
263 – Please remove the comment.
Ans: Removed. I’m not sure why it was included in the previous version.
288 – “p” in italics should be used consistently throughout the manuscript.
Ans: Adjusted. Thank you for the suggestion.
311 – “p-valor” – Please change to English. Also, review Table 2 and all tables to improve clarity and presentation for readers. For example, Table 1 spans two pages.
Ans: Adjusted. Thank you for your feedback. Regarding Table 1 crossing two pages, I tried multiple solutions, but even when placed on a single page, it extends beyond it. Reducing the font size resolves the issue but would conflict with the template guidelines.
371 – Please consider improving the figure quality.
Ans: I reinserted the original graph directly from the statistical software.
400 – Please review the table content and format. Apply this to all tables (1 to 5).
Ans: I attempted to revise the tables but was uncertain whether removing some information or using other variables would be better. I standardized all tables to align to the right, as requested by other reviewers, while retaining relevant details for each analysis.
The quality of writing in the discussion and subsequent sections could be improved.
Ans: I included two additional articles in the discussion and made some adjustments. I hope the revised information meets the expectations now.
All references should be corrected according to the journal template and instructions for authors.
Ans: Adjusted, with DOIs included as required by the journal.